# Measurement and outcomes of co-production in health and social care: a systematic review of empirical studies

Annika Nordin ![ORCID],[1] Sofia Kjellstrom ![ORCID],[1] Glenn Robert ![ORCID],[1,2] Daniel Masterson ![ORCID],[1] Kristina Areskoug Josefsson ![ORCID][1,3]

[1]Department of Quality improvement and Leadership, Jönköping Academy of Improvement of Health and Welfare, Jönköping University School of Health and Welfare, Jonkoping, Sweden
[2]Division of Methodologies, Florence Nightingale Faculty of Nursing, Midwifery & Palliative Care, King's College London, London, UK
[3]Department of Health Sciences, University West, Trollhattan, Sweden

**Correspondence to**
Annika Nordin;
annika.nordin@ju.se

## ABSTRACT

**Background** Co-production is promoted as an effective way of improving the quality of health and social care but the diversity of measures used in individual studies makes their outcomes difficult to interpret.

**Objective** The objective is to explore how empirical studies in health and social care have described the outcomes of co-production projects and how those outcomes were measured.

**Design and methods** A scoping review forms the basis for this systematic review. Search terms for the concepts (co-produc* OR coproduc* OR co-design* OR codesign*) and contexts (health OR 'public service* OR "public sector") were used in: CINAHL with Full Text (EBSCOHost), Cochrane Central Register of Controlled trials (Wiley), MEDLINE (EBSCOHost), PsycINFO (ProQuest), PubMed (legacy) and Scopus (Elsevier). There was no date limit. Papers describing the process, original data and outcomes of co-production were included. Protocols, reviews and theoretical, conceptual and psychometric papers were excluded. The Preferred Reporting Items for Systematic Reviews and Meta-Analyses guideline was followed. The Mixed Methods Appraisal Tool underpinned the quality of included papers.

**Results** 43 empirical studies were included. They were conducted in 12 countries, with the UK representing >50% of all papers. No paper was excluded due to the Mixed Methods Quality Appraisal screening and 60% of included papers were mixed methods studies. The extensive use of self-developed study-specific measures hampered comparisons and cumulative knowledge-building. Overall, the studies reported positive outcomes. Co-production was reported to be positively experienced and provided important learning.

**Conclusions** The lack of common approaches to measuring co-production is more problematic than the plurality of measurements itself. Co-production should be measured from three perspectives: outputs of co-production processes, the experiences of participating in co-production processes and outcomes of co-production. Both self-developed study-specific measures and established measures should be used. The maturity of this research field would benefit from the development and use of reporting guidelines.

### STRENGTHS AND LIMITATIONS OF THIS STUDY

⇒ This systematic review embraces co-production within both health and social care, which facilitates a holistic understanding of co-production from a user perspective.

⇒ This systematic review regards empirical studies, indicating that identified measures are relevant and possible to use practically, in empirical settings.

⇒ Only peer-reviewed research published in English is included, which may mean that important co-production research published in other languages has been overlooked.

## INTRODUCTION

Despite the term co-production stemming back to Ostrom's seminal work in the 1970s,[1] the study of co-production is nonetheless sometimes viewed today as having low scientific maturity.[2–4] For the purposes of this review—and following others[5]—we include co-design with its origins in the participatory design movement in Scandinavia also in the 1970s as a specific form of co-production. Co-production research has been reported as typically not outcome-focused, tending rather to describe co-production processes.[3 6] The concepts of co-production and co-design are attracting increasing research interest,[2 7] although while appearing to labour under a multitude of definitions.[7] This systematic review adopts the definition proposed by Osborne *et al*; 'the voluntary or involuntary involvement of public service users in any of the design, management, delivery and/or evaluation of public services'[5] (p. 640).

This systematic review is part of the Samskapa research programme on co-production[8] and concerns the measurement and outcomes of co-production as it relates to the provision of both health and social care services. Health and social care are provided by various organisational types in different countries, including private, public and

third sector organisations. Such services aim to improve social or health-related aspects of citizens' lives and are important for quality of life.[9] For example, the application of co-production in health and social care services is the creation, further development and evaluation of a smartphone app for young people with lived experience of self-harm.[10] To maximise the use and acceptance of the app, product users (young people with lived experiences of self-harm and clinicians), researchers and app developers collaborated through all stages of this project. The increasing number of such empirical studies of co-production in these sectors is predominantly in healthcare; of the 10 most cited papers on co-production, none concern social care.[2] Given widely held ambitions to integrate health and social care services,[9] this focus is problematic, and thus, this systematic review embraces both health and social care services.

Advocated as an effective way of improving quality of health and social care,[11–13] co-production is also considered more broadly as a means to promote a democratic ethos placing equal value on the contributions of both citizens and service providers in the successful delivery of services.[12 14] The benefits of patient and public involvement in research is also acknowledged as a moral obligation with several potential benefits.[15]

There are several plausible explanations related to the challenges researchers encounter when seeking to measure the outcomes of co-production,[16] including its emergent and adaptive nature which can impede the early determination of appropriate measurements and which might otherwise neglect outcomes that are significant to participants.[17] Co-production also often encompasses complex and broad needs,[18] and is characterised by both a multitude of stakeholders and a plurality of contexts; such features tend to lead to weak or indistinct cause-effect relationships between co-production activities and their outcomes.[19] However, knowledge of co-production outcomes is important in order to enable the continued improvement of services.[20] In summary, while important to explore the outcomes of co-production, these are often difficult to measure through a priori measures and deductive analysis,[6 21 22] and to date methodological diversity has been limited.[13]

Regardless of these challenges, scholars have previously used various categorisations to present outcomes, further complicating attempts to synthesise them and build a broader evidence base. Examples of categories of co-production outcomes include: outputs and outcomes,[11] discrete products, care processes and structural outcomes,[23] patient/staff involvement, generating ideas and tangible change[24] and both individual and collective outcomes.[4] Alternatively, innovation potential, individual well-being and citizen empowerment, increased effectiveness and efficiency, mobilisation of resources and increased democracy are categories suggested by Brix *et al*.[19] Other types of similar broader outcomes include increasing effectiveness, increasing citizen involvement, greater efficiency, gaining customer

satisfaction, strengthening social cohesion and democratising public services.[3]

There are a small number of reviews appraising the research designs and methods of co-production across the public sector. In their systematic review on empirical studies on co-production and co-design, Voorberg *et al* report that qualitative case studies, single or comparative, predominate.[3] More robust critiques of the designs and methods typically employed to study the outcomes of co-production may help to improve the scientific maturity of this research field and to enable better understanding of the wide range of reported outcomes. In this systematic review, outcomes are understood as qualitative and quantitative descriptions used to capture the impact, effect, results or outputs of ongoing, or implemented, co-production projects. Objectives, measures and outcomes are inter-related. Outcomes are described by the use of measures, and evaluated in relationship to stated objectives. In seeking to explore the measures and outcomes of co-production, knowledge of the project objectives is necessary.

The aim of this systematic review is to explore how empirical studies in health and social care have described the outcomes of co-production projects and how such outcomes have been measured.

The review questions are:
► What are the objectives of co-production projects?
► What measures have been used?
► What are the outcomes of co-production?

## MATERIALS AND METHODS

In the published Samskapa study protocol, the aims were to explore, enhance and measure the value of co-production for improving the health and social care of citizens.[8] A scoping review was performed as part of this wider study in March 2019,[7] and the included articles form the basis for this systematic review. This systematic review has followed the Preferred Reporting Items for Systematic Reviews and Meta-Analyses (PRISMA) guidelines,[25] and the manuscript is organised according to the instructions for authors.

### Information sources, eligibility criteria, search strategy and selection process for the scoping review

The scoping review explores 'what is out there' in the peer-reviewed research relevant to co-production or co-design in health and social care services.[7] The full search strategy is provided in online supplemental file 1. In brief, search terms for the concepts (co-produc* OR coproduc* OR co-design* OR codesign*) and contexts (health OR 'public service* OR "public sector") were used in the following databases: CINAHL with Full Text (EBSCOHost), Cochrane Central Register of Controlled trials (Wiley), MEDLINE (EBSCOHost), PsycINFO (ProQuest), PubMed (legacy) and Scopus (Elsevier). There was no specified date limit. Articles were included in the scoping review if they related to peer-reviewed research, consisted of any methodology relevant to

co-production or co-design in the context of health and social care services, involving service-users and were written in English. The results were stored in a database in Excel and in Rayyan, a web-based app and mobile app for systematic reviews that help expedite the initial screening of abstracts and titles using a semi-automation.[26]

### Eligibility criteria and selection process for this systematic review

The 979 records from the scoping review were screened initially by one of the researchers (DM), to remove papers without empirical data or explicit references to measurements. The titles and abstracts of identified papers were added into a new Rayyan data base and reviewed by two of the researchers (AN and KAJ), who independently noted 'include', 'unsure' or 'exclude' in Rayyan based on the three additional eligibility criteria for this systematic review:

1. Process of co-production is described (inclusion criterion).
2. Original data from process and outcomes is described (inclusion criterion).
3. Not a theoretical, conceptual, protocol, review or psychometric paper (exclusion criterion).

Conflicting decisions, or decisions the researchers were indecisive of, were crosschecked and validated by the two researchers and remaining conflicts were discussed until consensus was reached. The eligibility criteria were then applied on the included full-text papers. To enhance the reliability of the review process, the two researchers read the same first five papers and discussed their decisions until consensus was reached. Another five papers were reviewed following the same procedure and thereafter, remaining papers were examined separately, with decisions noted in a shared Excel file. The quality of each included papers was then appraised.

The Mixed Methods quality Appraisal Tool (MMAT) has been developed to assess the methodological quality of empirical studies[27] and was used to appraise the quality of included full-text papers. MMAT includes five study designs (qualitative, randomised controlled trial, non-randomised, quantitative descriptive and mixed methods) and for each study design, five criteria for the evaluation of the methodological quality are provided.[28] The criteria can be evaluated 'yes', 'no' or 'cannot tell', and affirmative responses indicate quality. Two initial screening questions

**Table 2** Variables for study characteristics and data extraction definitions

| Study characteristics | Data extraction definitions |
| --- | --- |
| Year | Year of publication |
| Country | Country in which the study was conducted |
| Field | The specific healthcare or social care field for the study |

are included to assess whether the study is empirical. The first five papers were assessed jointly by two researchers (AN and KAJ). Another five papers were assessed separately and the results compared and discussed. On the basis of the MMAT tool, the remaining papers were assessed independently and the results documented in a shared Excel file. Two meetings of the researchers were held to discuss queries. The MMAT study designs are used as categories to group the included studies in the synthesis of our findings below.

### Data collection process and data items for this systematic review

To develop a shared work procedure for the data collection, two researchers (AN and KAJ) jointly extracted data from five papers and compared and discussed the results. The remaining data extraction was carried out independently by the two researchers and documented in separate files, one for each MMAT category. To enhance consistency, data extraction definitions were inserted as column headings in the Excel files. A count of the number of papers within each MMAT study design category was made. A count was also completed for each of the study design characteristic variables and the proportion of affirmative responses on the MMAT screening questions. After completion of the data extraction, the findings were discussed with PhD students in the Samskapa research programme. The discussion was intended to provide a fuller understanding of the meaning of the results for our future work in the wider programme. A plain language blogpost including discussion points inviting comments from a wider audience was also written.[29]

In table 1, the research questions are presented alongside the definitions for data extraction.

**Table 1** Research questions and data extraction definitions

| Research questions | Data extraction definitions |
| --- | --- |
| What are the objectives for co-production projects? | Descriptions of the initiators of the projects and the aims of the studies. |
| What measures are used? | Descriptions of used measures (scales, questionnaires or other forms of measurements). |
| What outcomes of co-production are reported? | Qualitative or quantitative descriptions of the impact, effects, results or outputs of ongoing or implemented co-production projects, including learning and social and cultural effects. |

In table 2, other variables for which data were sought are listed and defined.

## Synthesis methods

The extracted data in each MMAT study design category was analysed separately. In the first step of analysis, the data extractions were closely read. In the next step, the data extractions formed the basis for a convergent qualitative synthesis.[28] A synthesis is a process of putting the findings from individual studies together into a new or different arrangement and thus providing knowledge that would not otherwise be apparent while reading the individual studies in isolation.[30] The measures were analysed and coded into four categories: (1) international validated scales (including scales with minor adaption or use of subscales), (2) measures based on publications (eg, based on public documents, published research, indicators and non-validated scales), (3) self-developed study-specific measures (locally developed measures) and (4) measures on the experience of co-production.

## Patient and public involvement

Patients and/or the public were not involved in the design, or conduct, or reporting, or dissemination plans of this research.

## RESULTS OF THE SYSTEMATIC REVIEW
## Study selection

The first initial screening of all papers included in the scoping review reduced the number of potential papers from 979 to 187. Thereafter, the titles and abstracts of the remaining papers were reviewed and 116 papers were excluded. Thirty-seven papers were excluded based on the first eligibility criteria, 34 papers were based on the second eligibility criteria and 45 were based on the third eligibility criteria. The eligibility criteria were then applied on the included full-text papers, leading to the exclusion of a further 28 papers. Fifteen papers were excluded based on the first eligibility criteria, eight papers based on the second eligibility criteria and five based on the third eligibility criteria. The remaining 43 papers formed the basis for the quality appraisal and the data extraction for this systematic review. The PRISMA 2020 flow diagram for systematic reviews is presented in figure 1.[31]

## Result of the quality assessment

Of all included papers in the systematic review, 86% were considered as having a clear research question (screening question no. 1 of MMAT) and 95% to have a relevant data collection process (screening question no. 2 of MMAT). The largest proportion of included papers (60%) reported on studies with mixed methods study designs (table 3). The distribution of affirmative responses to the MMAT categories indicate the general high quality of the included studies.

## Study characteristics

The studies located in this review were published between 2012 and 2019 (online supplemental figure 1). The year 2018 saw the most papers (n=17), however only the first 3 months of 2019 are included.

The 43 studies were conducted in 12 different countries, generally with three or fewer studies per country. The exceptions are the UK (n=23) and Australia (n=6). The studies were conducted in different fields of health and social care; mental health has the largest proportion of studies (n=14). The UK represents 53% of the included studies and of the 23 studies conducted there, 9 related to a mental health setting.

Based on our convergent design,[28] our findings are reported below for each research question in turn. In the text, user is used as a generic term for those who are using

**Table 3** Proportion of affirmative categories in MMAT per study design

| | Yes on 5 MMAT criteria | Yes on 4 MMAT criteria | Yes on 3 MMAT criteria | Yes on 2 MMAT criteria | Yes on 1 MMAT criteria | Yes on 0 MMAT criteria |
|---|---|---|---|---|---|---|
| **Qualitative studies** 12% (5) | 60% (3) | 0% (0) | 40% (2) | 0% (0) | 0% (0) | 0% (0) |
| **Quantitative randomised controlled trials** 7% (4) | 25% (1) | 0% (0) | 25% (1) | 25% (1) | 25% (1) | 0% (0) |
| **Quantitative non-randomised studies** 7% (3) | 0% (0) | 33.3% (1) | 33.3% (1) | 0% (0) | 0% (0) | 33.3% (1) |
| **Quantitative descriptive studies** 14% (6) | 50% (3) | 33% (2) | 0% (0) | 17% (1) | 0% (0) | 0% (0) |
| **Mixed methods studies** 60% (25) | 40% (10) | 24% (6) | 16% (4) | 12% (3) | 8% (2) | 0% (0) |

MMAT, Mixed Methods quality Appraisal Tool.

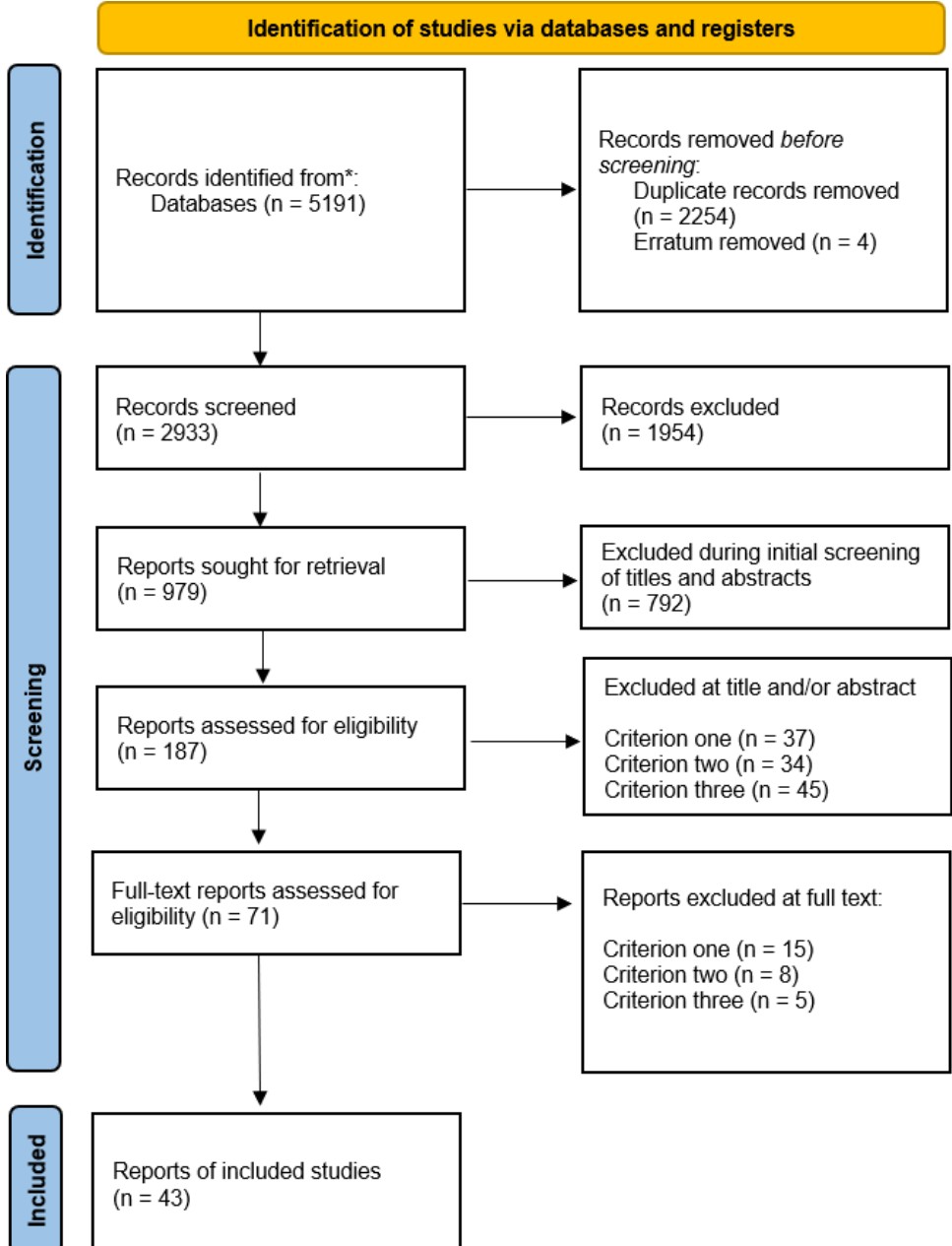

**Figure 1** The Preferred Reporting Items for Systematic Reviews and Meta-Analyses flow diagram online supplemental file 1, full search strategy.

a service, innovation or healthcare innovation in health and social care. The term informal caregiver is used as a generic term for individuals supporting and helping the users, based on personal engagement (and not a professional responsibility) and providers is used as a generic term for those who are providing a service. The outcomes and measures are compiled in online supplemental file 2.

### Results of syntheses

#### What are the objectives of co-production projects?

The objectives were explored by analysis of who initiated the co-production projects and establishing the aims of the study. Users rarely initiated co-production projects or studies.

The aims of the *qualitative studies* ranged from increasing communication with users to encouraging service innovation in partnership with them. Some of the studies also aimed to increase users' engagement with health and social care services and to assess how users and providers perceive services and experience quality.[32] In the few cases, the origins of the studies were described, the studies had been initiated by researchers.

All the *quantitative randomised controlled trials* aimed to evaluate outcomes of co-produced interventions. Two studies aimed to provide better, more timely support through the use of digital tools.[10 33] One study aimed to increase the patient and public involvement in research[34]

and a further study aimed at improving staff behaviours and actions to improve care through co-produced interventions.[35] Those who initiated the projects—and their reasons for doing so—were not described in the studies.

Descriptions of the initiators of project were only included in two of the three *quantitative non-randomised controlled studies*. The project by Hahn-Goldberg *et al*[36] was initiated by the hospital management after a co-produced pilot study and the project by Bolton *et al*[37] was initiated by several organisations aiming to promote public health. All three studies aimed at improving users' experiences of provided services.[36–38]

None of the included *quantitative descriptive studies* described users as having initiated the projects. Chapman *et al*[39] aimed at evaluating a co-produced plan to reduce emergency visits. Lamph *et al*[40] aimed at evaluating co-production in an e-learning programme to improve staff attitudes. Clark *et al*[41] aimed at evaluating a co-designed and co-produced programme to improve patients' health and self-management skills and Roberts *et al*[42] aimed at improving care through individualised care quality data. Some studies aimed at increasing co-production within processes. Cramm and Nieboer[43] aimed at developing the care process by improving the co-productive relationship. Adinolfi *et al*[18] aimed to improve co-production processes and patient outcomes by integrating health and social care and increasing user empowerment.

The aims of the *mixed methods studies* varied. One study aimed to empower disadvantaged communities[44] and one aimed at testing co-production in areas with high levels of violence.[45] Other aims were to evaluate the experience of partaking in co-design[46] and to evaluate how co-design can improve patient experiences and services.[47] Four studies aimed at improving co-production within already ongoing work processes.[48–51] Most of the studies aimed at evaluating outcomes of co-produced information, interventions/services or products.[47 48 52–67] Among the studies evaluating outcomes of co-produced interventions/services, the focus was on effectiveness[52 61] and on the level of improvement.[54] Evaluation of acceptability and feasibility of co-produced interventions or products was the aim of several studies[58 65 67] and two studies aimed at evaluating the use and experience of a co-designed product.[59 60] Other studies aimed at evaluating user-led co-produced interventions.[63 66] However, of the 25 mixed methods studies only 1 stated that the study was initiated by users.[56] Also, only one study compared a co-produced method with other methods.[68]

## What measures are used?

In the *qualitative studies*, no measures beyond simple descriptive calculations were used.

In the *quantitative randomised controlled trials* several validated scales were used. Two studies used three different validated scales each[10 35] and one[33] used two validated scales. The scales focused on health or clinical aspects related to the interventions. Two self-developed study-specific measurements purposing to evaluate the

intervention itself were used[10 34 35] and in the study by Hastings *et al*[35] a scale based on previous publications was used.

Self-developed study-specific measurements focusing on the experiences of an intervention and the implementation process were used in the *quantitative non-randomised controlled* studies.[36–38] In the study by Bolton *et al*,[37] two validated scales focusing on health aspects related to the intervention and three questionnaires based on other publications were also used. It is worth noting that the validity and reliability of the outcome measurements were rarely discussed. The focus was more on the sustainability and usefulness of the innovations/interventions.

In three of the six *quantitative descriptive studies* validated scales were used. Clark *et al*[41] used five different validated scales. Cramm and Nieboer[43] used two validated scales and Adinolfi *et al*[18] used one validated scale. The scales focused on health or clinical aspects related to the interventions. Two studies included self-developed study-specific measurements concerning the interventions[18 39] and one study included self-developed measurements regarding the clinical outcomes.[42] Three studies used measurements based on previous publications.[18 40 42] Cramm and Nieboer[43] used The Relational Coordination Scale (RCS) to assess participants' perceptions of interaction productivity, thus measuring the experience of the co-production process itself.

Seven of the 25 *mixed methods studies* used validated scales. In three studies, one validated scale was used.[51 55 67] Three studies used two different validated scales[48 58 65] and Ferguson *et al*[49] used three different validated scales. Two of the studies used scales that, unlike the others, did not focus on health or clinical aspects. Instead, they used validated scales measuring the quality[55] and usability of technical systems or innovations.[51] Eleven studies used one or more measures based on previous research.[45 47 52–58 61 67] Five studies did not use any self-developed study-specific measures[47 48 56 57 67]; all others did. Two studies used measures regarding the experience or level of the co-production process. Revenäs *et al*[46] used a self-developed survey to assess users' experience of participating in the co-production process, and Haynes *et al*[56] measured how the intervention adhered to the National Health and Medical Research Council's (NHMRC) principles. Brosseau *et al*[50] did not measure the level of co-production but referred to several ways this could have been done. Morales-Perez *et al*[45] also did not measure the co-production level; however, they concluded that the study results were made possible due to the high levels of participatory engagement.

## What outcomes of co-production are reported?

A wide variety of outcomes were presented in the *qualitative studies*, commonly describing the number of innovations or implemented changes, the establishment of priority areas or consensual agreement and the development of a tool.[32 69–72] The sustainability of outcomes was presented as having to encompass financial sustainability,

new organisational relationships and further quantitative studies.[70] Organisational structures and issues related to patient safety were described as potentially hampering the implementation of digital tools, thus influencing sustainability negatively.[32]

The outcomes in the *quantitative randomised controlled trials* were positive for two interventions targeting users, showing significant reduced self-harm and reduced mental ill-health, and that the users wanted to continue using the co-produced digital tools.[10 33] A third study targeting mental health patients did not report any positive outcomes.[34] The outcomes for interventions targeting providers were also favourable, but the results were not as strong as in the studies targeting users.[35]

Regarding the *quantitative non-randomised controlled studies*, Hahn-Goldberg *et al*[36] presented outcomes related to the use of a new tool, but the effects of the tool, including the costs are yet to be evaluated. With the objective of identifying the vital outcomes of the project, Murphy *et al*[38] used several measurements covering different perspectives. The measurements concerned outcomes of the process itself, user surveys and user narratives. Some of the surveys aimed at family members had response rates too low to provide useful data. However, narratives from users were used to complement the survey data and these data showed both positive and sustainable outcomes. The research teams concluded that the value of collaborating with stakeholders to find useful measurement and tools for assessments was an important learning outcome. Bolton *et al*[37] used measures not tested beyond their face validity but there were other outcomes showing positive results such as new social support networks and user requests for support in gaining knowledge supporting their health.

Cramm and Nieboer[43] and Lamph *et al*[40] described quantitative results and correlations from established questionnaires in their *quantitative descriptive studies*. However, Lamph *et al*[40] also included observational process outcomes. Clark *et al*[41] additionally included health data, registry data and observational process outcomes. Chapman *et al*[39] described quantitative results of objective measurements and Roberts *et al*[42] described compliance with clinical practices and quality indicators. Quantitative outcomes in terms of individual health status and cost savings on organisational levels were also reported by Adinolfi *et al*.[18]

The outcomes in the *mixed methods studies* were extensive and to facilitate an overview the outcomes are presented below in four broad themes: effects on the outcomes and processes, success factors, impact on participants and learning outcomes.

### Outcomes and processes

Co-production led to many effects on the outcomes and processes the projects aimed to improve, with improved care,[62] stronger social networks[63] and increased urgency in communication as[56] some examples. Co-production with users indicated what worked well and what needed

to be improved[48 66] and combining quality improvement and co-design led to additional quality improvement projects.[50] Co-produced products like reusable learning objectives were highly cost-effective and showed significant effects on outcomes, knowledge, skills, uptake and adherence. Reusable learning objectives were also highly rated as useful and recommended to others.[49] Studies measuring outcomes during the development of a technical product reported that co-production with users was helpful, and that it provided understanding valuable for the further improvement of the product.[55 59 67] In one study the product uncovered unmet health needs among participants[55] and in another study, the final testing revealed that a co-produced product had led to improved symptoms.[60] One study resulted in a non-functioning IT product.[51]

There were positive effects of using co-production to improve processes,[54 55 57 58 65] and the combination of quantitative and qualitative measurements led to a greater understanding of the components influencing the results[54 58 65] and additional increased engagement by families.[54] The results also specified requirements for further development of tested interventions.[58] D'Young *et al*[61] reported that co-production lead to the fulfilment of the effectiveness goals of a service.

### Success factors

Close engagement and co-production with those intended to take part in, or benefitting from the intervention, were identified in several *mixed methods studies* as success factors.[44 45 48] Another identified success factor was the use of emancipatory process indicators (eg, the use of an emoticon survey on important values) in the implementation and evaluation of projects[56] and facilitator training.[45]

### Impact on participants

Co-production was an influential experience for participants and organisations.[45 48 52 53 66] Improved well-being,[63] respect and confidence,[45] and empowerment[56] were examples of influential experiences. However, participation in co-production projects also lead to paradoxical experiences as a wish to engage more participants, while preferring to work in separate groups: a desire for more preparation and discussion, while considering invested time as a concern; and the view on co-design as valuable for improved care, while doubting the realisation of co-care in practice.[46] The participation also led to feeling of uncertainty and vulnerability in one study.[63] In their study, Revenäs *et al*[46] suggested that co-design methods influenced the participants' experiences of the process, thus concluding that such methods need to be adjusted to participating stakeholders and contexts.

### Learning outcomes

Co-production led to several unforeseen learning outcomes and learned principles were highly valuable in the next step of improvement and development.[56] One study resulted in learning outcomes such as the

importance of motivation; identification of most functional techniques; evaluation of mock-ups with end-users and the recognition of the influence of informal caregivers.[51] Similar learning outcomes were also presented as results in other studies. Rosso and McGrath[44] summarised the learning outcomes of their study as the importance of: formatting a regional partnership action group; selecting appropriate communities according to needs and preparedness; engaging community champions in each location; establishing a continuous consultation framework relating to each location and the regional action group; co-designing culturally appropriate age-relevant activities in collaboration with local stakeholders and considering sustainability. Boyd *et al*[47] summarised the learning outcomes of their study as the value of using co-design alongside traditional quality improvement methodologies; early engagement with patients; staff buy-in and individuals trying things outside their comfort zones. Another study brought forward methodological learning outcomes, that is, the synergy between lean and experience-based co-design.[50] Only two studies with a mixed methods design measured sustainability over time.[52 64] Notable, the only study comparing co-production with other methods stated that the data gathered (in this case, patient incident reports) was unlikely to have been found through other established methods.[68]

## ANALYSIS AND DISCUSSION
### General interpretations of results
Knowledge of the outcomes of co-production is fundamental for assessing the value of co-production projects, and to inform approaches to continually improve services important to citizens. Overall, the studies included in this systematic review showed positive outcomes from co-production but these were difficult to compare across the studies. One plausible explanation for this lies in the ambiguous nature of the various ways in which co-production is conceptualised. Co-production is, and has been, defined and applied in multiple ways in health and social care.[7] Nonetheless, co-production was a positive experience for participants in our included studies and the findings provided learning important to enable a range of stakeholders to continuously engage in such processes. Various measures were used, and the outcomes were presented in a diversity of categorisations. The results of this systematic review point at three perspectives that are important for measuring the outcomes of co-production. These perspectives, and observations regarding study designs are discussed below.

### Perspective 1: outputs of co-production processes
The first perspective concerns outputs. Outputs are the results of the processes that are undertaken to contribute to the stated objectives of a project.[73] When outputs are measured continuously, they can also be used to monitor ongoing work.[74] The extensive use of a range of self-developed study-specific measurements in the included

studies limited comparisons and cumulative knowledge-building regarding co-production. Two exceptions were found where the validated measures concerned the quality or usability of the intervention itself, that is, outputs of the co-production process. The Mobile Application Rating Scale was used to explore users' acceptance and experience of a co-produced app[55] and the System Usability Scale was used to assess user experience of a mobile app.[51]

A precondition for identifying relevant measures in relation to outputs is that the processes are carefully described, and a fundamental aspect of co-production processes concerns how and when users are involved. In general, these processes have been described in detail; however, the clarity concerning in which phases of the process users participated could have been clearer in many of the studies. Without this, studies appear to have been co-produced even when participation was restricted to only one phase. In their generic programme theory for co-production, Brix *et al*[19] suggest three phases (co-design/co-planning, co-delivery/co-creation and co-review/co-evaluation) which may be helpful to clarify when and how users participate.

### Perspective 2: experiences of participating in co-production processes
The second perspective concerns participants' experiences of participating in co-production. Given that co-production is anchored in democratic and equal value principles,[12 14] it is notable that only three studies provided insights into how these core values might be measured. The RCS was used to assess how participants perceived the interaction among actors during the collaboration[43] and the NHMRC's principles guided a strong focus on values important for collaboration with indigenous communities.[56] The self-developed questionnaire by Revenäs *et al*[46] measured how participants experienced the possibility of having their voice heard during a co-production collaboration. In general, users' objectives to participate were not well described. As their participation is fundamental for co-production, this is notable. If the initiative takers are unaware of the users' intrinsic and extrinsic aims for participation, they can neither take them into account nor create receptive conditions for them to contribute. There is a need to take users' objectives for participating into explicit account, and to measure experiences of participation.

### Perspective 3: outcomes of co-production
The third perspective concerns the interventions' stated objectives and expected outcomes.[11] Twenty-six of the 28 validated scales in this systematic review involved this perspective and covered different aspects of improved health or better clinical outcomes. Overall, the studies reported positive outcomes. However, there is a difference between positive outcomes and goal fulfilment. Various kinds of measures were used but the goal levels for them were seldom set before projects started. Users

were seldom involved in the choice of measures or decisions on goal levels. This lack of shared operational objectives makes it difficult to evaluate if the co-production projects reached their expected outcomes—especially from the perspective of users—and has to our knowledge been less problematised in the literature than the lack of shared measurements per se.

### Domination of mixed methods studies

Mixed methods study designs are well-suited for studying complex phenomena such as changes in behaviours or experiences from different perspectives[75]; 60% of the studies in this systematic review used different mixed methods study designs. The use of mixed methods study design in research in health and social care is increasing,[76] but qualitative study designs are typically the most common study designs in co-production research.[77] In qualitative studies, measurements do not have the same central role as in quantitative studies and in the mixed methods studies included in this systematic review, qualitative methods were prominent. The study designs identified in this systematic review are in line with both the general trend towards more mixed study designs in health and social care, and the use of qualitative approaches in co-production.

### Implications for practice

In this systematic review, we did not identify any generic measures of the outcomes of co-production and nor would we recommend such an approach. Instead, we suggest a model facilitating comparisons between studies and projects that does not limit flexibility and can incorporate the use of both self-developed study-specific measures and established measures. In complex contexts, cause-and-effect relationships are not linear and unexpected consequences are common. To address this, a process for selecting a small number of measures representing different perspectives has been suggested.[78 79] Balanced score card[80] and the value compass[78] are examples of models that seek to help pinpoint different perspectives which are important for measuring and monitoring organisational performance and processes.

We propose a model for balanced measurement of co-production (figure 2), which combines output measures (perspective 1) and outcome measures (perspective 3). Our emphasis on continuous output measures is in line with a continuous quality improvement approach since this clarifies *how* the processes or interventions works. Both perspectives 1 and 3 are important to monitor and evaluate change.[79] The model also aligns with the core values of co-production by incorporating how participants experience their own participation in the co-production process (perspective 2). The model proposes the use of at least one measure per perspective, and the use of both established and self-developed study-specific measures. The chosen measures should be selected to offer actionable information for further improvements.

The notion of 'balanced' in this context refers to the act of weighing the outcomes together. A balanced approach may prevent co-production from being used to achieve project objectives without addressing its democratic principles. It may also help focus attention on co-production projects in which participants report positive experiences but without the project objectives being fulfilled. Based on how we have found co-production to have been reported in the 43 studies in this systematic review, the model is a radical proposition; only Adinolfi *et al*[18] have measures reflecting all three proposed perspectives.

The model also emphasises the use of both established measures (online supplemental file 2, appendix 1, first two columns) and self-developed study-specific measures (online supplemental file 2, appendix 1, third column). Established measures (internationally validated scales and measures based on publications) are depicted as a light-grey core at the centre of the model. This core of measurements corresponds with the outcome measures suggested by Marsilio *et al*,[77] for example, validated scales, other metrics and indicators. In our systematic review, a multitude of measures based on publications were used whereas the use of internationally validated scales was much more restricted. This may indicate a lack of knowledge of existing validated scales. Broadened multidisciplinary competence among researchers and professionals

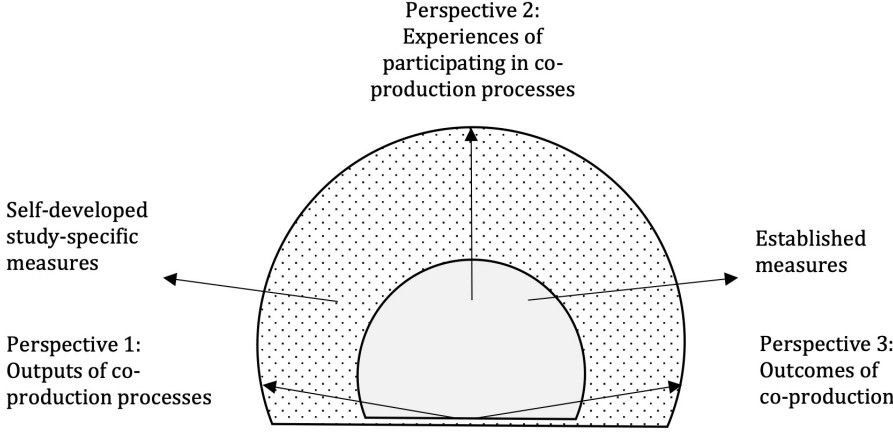

**Figure 2** A model for the balanced measurement of co-production.

in the field of co-production may support a greater use of validated scales, which could then strengthen the reliability of results and promote comparisons. The model also includes self-developed study-specific measures, placed in the dotted zone surrounding the core (figure 2). Such measures are 'tailor-made' for the specific purposes of a single study and their joint development by users and providers within a study can be a foundation for shared understanding, creativity and engagement.

Our proposal for self-developed study-specific measures is in line with the findings in the systematic review by Greenhalgh *et al* regarding frameworks for patient and public involvement,[81] in which they state that it might be more efficient and sustainable to develop one's own framework than to apply someone else's. Another implication from the review by Greenhalgh *et al* relating to our suggested model is the need for the actors themselves to formulate operational objectives, so that they—with their own measures—can determine whether the co-production project has reached its intended goals. Finally, included papers in this systematic review generally showed a low awareness regarding the difference between outputs and outcomes, and we believe that the proposed model can clarify both the differences between the two perspectives, and their shared importance.

### Implications for future research

An important implication for research concerns the need for closely described co-production processes, including clarifications of the activities, or process steps, that users participate in. Such clarity would provide better understanding of how co-production activities impact outcomes. A good example of how to make the co-production process in research transparent is provided by Marks *et al*.[82] In their scoping review, they list specific citizen science activities and group them into categories describing increasing public involvement (contributory, collaborative, co-created and citizen-led).

A second implication for future research concerns the composition of project groups, which in general need to include more multidisciplinary competences. A broader knowledge base can facilitate the identification and use of established measures, thus linking local co-production projects to a cumulative body of knowledge. A third implication concerns the description of measures. In several papers, authors refer to measures, surveys or survey questions without describing them. Regardless of the type of measures, these should be described in sufficient detail that researchers can consider them for other studies. A fourth implication concerns the need for better knowledge of which mixed methods study designs are most suited for research in co-production in health and welfare services. Lastly, the diversity in how co-production research was reported in the papers included in this systematic review points to a major implication, the development of reporting guidelines for co-production research. The adoption and use of such a guideline would significantly contribute to the scientific maturity of the research field.

### Strengths and limitations

This review is in line with the published study protocol[8] and contributes to the aim of exploring, enhancing and measuring the value of co-production for improving the health and social care of citizens. Against the background of the widespread ambitions to integrate health and social care services, while the number of empirical studies on co-production is increasing in healthcare, it is a strength that this systematic review includes both health and social care services. Furthermore, the MMAT tool was crucial for appraising the quality of included studies and provided an overview of the used study designs. The categories used to code and analyse the extracted measures were also easy to use and can be recommended to other studies and projects on co-production. The identified measures are empirically tested which indicates that they are relevant and possible to use in other empirical settings.

The data for this systematic review were obtained from a scoping review, in which only peer-reviewed research, published in English were included.[7] Although justified considering the aims of this systematic review, this inherently narrowed our focus and may mean we have missed important applied co-production research published in other languages. We also acknowledge that it would be ideal to update the review. However, the detailed analysis was a time-consuming process and repeating the analysis would create a similar delay between an updated search and publication. Furthermore, research exploring Cochrane reviews described that only a minority were updated to incorporate evidence from new primary studies. Of those that did, very few updates led to changes in the conclusions.[83] We believe this applies to our systematic review as well and that the overall findings are most unlikely to change with an update covering only a small number of years. The more recently published reviews with which we have compared our results lend weight to this belief.

### Conclusions

The objective of this systematic review was to increase understanding of how empirical studies have measured the outcomes of co-production and co-design in health and social care and what the outcomes are. Overall, the studies reported positive outcomes using a multitude of measures. This plurality limited comparisons and cumulative knowledge building but the lack of common strategies and models for measuring co-production is more problematic than the plurality itself. The findings of this systematic review suggest that co-production should be measured from three perspectives: the outputs of co-production processes, the experiences of participating in co-production processes and outcomes of co-production. Two types of measures should be used: self-developed study-specific measures and established measures. By measuring and monitoring co-production with this balanced approach, the democratic aspects of co-production and its complexity can be incorporated within future

evaluations. By using the model for the balanced measurement of co-production, we believe that future studies in co-production can better contribute to 'the science of participation'.[84] The development of a reporting guideline would also enhance the needed transparency and the scientific maturity of this research field.

**Acknowledgements** We are grateful for the help from all participants in co-production events, and from our university library. We also thank our colleague Bertil Lindenfalk, who was engaged in the improvement of figure 2.

**Contributors** All authors took part in the study conception and design. AN, DM and KAJ performed the data collection. AN and KAJ performed the data extraction, analysis and interpretation of results. AN and KAJ made the draft manuscript preparation. DM and AN wrote the 'Materials and methods' section and AN wrote the 'Discussion' section. All authors reviewed the results and approved the final version of the manuscript and accepted to be accountable for all aspects of the work.

**Funding** This work is part of the Samskapa program which is supported by Forte: Forskningsrådet om Hälsa, Arbetsliv och Välfärd, grant number 2018-01431.

**Competing interests** None declared.

**Patient and public involvement** Patients and/or the public were not involved in the design, or conduct, or reporting, or dissemination plans of this research.

**Patient consent for publication** Not applicable.

**Ethics approval** Not applicable.

**Provenance and peer review** Not commissioned; externally peer reviewed.

**Data availability statement** Data are available on reasonable request.

**ORCID iDs**
Annika Nordin http://orcid.org/0000-0002-2480-1641
Sofia Kjellstrom http://orcid.org/0000-00018952-8773
Glenn Robert http://orcid.org/0000-0001-8781-6675
Daniel Masterson http://orcid.org/0000-00034364-9814
Kristina Areskoug Josefsson http://orcid.org/0000-0002-7669-4702

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
