## [Reviewer comments · BMJ Open]

ARTICLE DETAILS

TITLE (PROVISIONAL)	The measurement and outcomes of co-production in health and social care - a systematic review of empirical studies
AUTHORS	Nordin, Annika; Kjellstrom, Sofia; Robert, Glenn; Masterson, Daniel; Areskoug Josefsson, Kristina

VERSION 1 – REVIEW

REVIEWER	Langley, Joe Sheffield Hallam University, Lab4Living
REVIEW RETURNED	13-Apr-2023

GENERAL COMMENTS	Thanks for the opportunity to review this article summarising your extensive piece of work. The article is very well written, and informative in my opinion. A valuable contribution. I had very little to identify to improve it - a couple of minor spelling and grammar points and a recommendation to revise your balance model image. Perhaps through the black and white re-production, I could only see black or grey semi-circles and some arrows. No labels or other visual information to enable me to understand what the model itself was all about (aside from your text description in the manuscript). The model image needs some work. Thanks
---

REVIEWER	Ansell, Margaret University of Florida, Health Sciences Center Libraries
REVIEW RETURNED	25-Apr-2023

GENERAL COMMENTS	The abstract doesn't mention the review's inclusion/exclusion criteria, which is generally standard for reporting systematic reviews. It also doesn't specify how this systematic review is more focused in scope than the original scoping review search upon which it is based. In the introduction, it would be helpful context for the reader to understand the application of the concept of incorporating co-production into health service research, design, delivery, and evaluation. The search process described includes some errors: the peer-review filter in EBSCO only restricts to journals that conduct peer review - columns, opinion pieces, and letters to the editor from journals that also publish peer-reviewed articles are included in search results with that filter activated. The description should be altered to "Articles were included in the scoping review if they were published in journals that conduct peer-review...". Additionally, on
---

	Page 5, Line 11, "involving service-users" is not quite correct - while "public service" and "public sector" were used as search terms, they were OR'd with terms related to health, so health care services that don't involve service-users could have been included in results as well. The inclusion/exclusion criteria reported were that of the prior scoping review, rather than that of the systematic review based on those original search results. If the presence of empirical data and measurements were required, as the 'initial screening' step would imply that they are, then they should be listed as inclusion criteria alongside the other inclusion criteria. It is also concerning that this step was taken separately from the following title/abstract screening step, which was conducted by two researchers and included additional checks to avoid bias. In general, the authors need to clarify and distinguish the methods of the original scoping review from this systematic review. If the MMAT was used to assess quality, why was quality also assessed through the researchers discussing the data collection methodology and examples of the first findings with PhD students in the Samskapa research program? How did that impact the results? There doesn't seem to have been any significant quality assessment conducted. While the limitations mention the need for an update, they don't clarify whether the original scoping review search should be updated, or the subsection utilized for the systematic review.
--	--

REVIEWER	Tuurnas, Sanna University of Vaasa
REVIEW RETURNED	26-May-2023

GENERAL COMMENTS	The study is well conducted and addresses an important question in field of co-production: the measurement of outcomes. The results show that there is a lack of common approaches to measuring outcomes of co-production. In my view this has a lot to do with the ambiguous nature of co-production concept itself. In this respect I was missing some critical discussion about the pluralism of the co-production concept and its impact on the outcomes or the results of study. As is well known, co-production can mean very different things from actual co-production to co-design. Co-production also takes place between different types of actors and agency (managers, professionals, politicians, patients, service users, citizens, family members). My comment therefore is, how are the type of co-production and the types of agency taken into an account in the research and how does this impact the way the process is measured, and what the outcomes are? Otherwise, I think that the manuscript is mainly ready for publication. Overall, a very interesting study!
---

VERSION 1 – AUTHOR RESPONSE

Reviewer Dr. Joe Langley, Sheffield Hallam University	
1. A couple of minor spelling and grammar points	Suggested changes are implemented.

2. Revise your balance model image. Perhaps through the black and white re-production, I could only see black or grey semi-circles and some arrows. No labels or other visual information to enable me to understand what the model itself was all about (aside from your text description in the manuscript). The model image needs some work	The model is revised so that the design of the full model, including labels are visible when uploaded.
Reviewer Ms. Margaret Ansell, University of Florida	
3. The abstract doesn't mention the review's inclusion/exclusion criteria, which is generally standard for reporting systematic reviews.	Abstract is updated with three sentences. ("There was no date limit. Papers describing the process, original data and outcomes of co-production were included. Protocols, reviews and theoretical, conceptual and psychometric papers were excluded").
4. The abstract also doesn't specify how this systematic review is more focused in scope than the original scoping review search upon which it is based.	Abstract is updated. Information regarding the specific inclusion/exclusion criteria is added (see No 7 above). Furthermore, the Strengths and limitations section states that "This systematic review regards empirical studies, indicating that identified measures are relevant and possible to use practically, in empirical settings".
5. In the introduction, it would be helpful context for the reader to understand the application of the concept of incorporating co-production into health service research, design, delivery, and evaluation.	Two sentences exemplifying the practical application of co-production in health and social care have been included in the Introduction. ("An example, the application of co-production in health and social care services is the creation, further development and evaluation of a smartphone app for young people with lived experience of self-harm [10]. To maximize the use and acceptance of the app, product users (young people with lived experiences of self-harm and clinicians), researchers and app developers collaborated through all stages of this project").
6. The search process described includes some errors: the peer-review filter in EBSCO only restricts to journals that conduct peer review - columns, opinion pieces, and letters to the editor from journals that also publish peer-reviewed articles are included in search results with that filter activated. The description should be altered to "Articles were included in the scoping review if they were published in journals that conduct peer-review...".	This was not determined by the search terms or databases but by the team reviewing these articles. These were manually removed using the exclusion criterion during the scoping review. This is explained in closer detail in the scoping review which is cited in the systematic review. To further clarify this, the reference is also added in the Method section. (Masterson et al., 2022).
7. Additionally, on Page 5, Line 11, "involving service-users" is not quite correct - while "public service" and "public sector" were used as search terms, they were OR'd with terms related to health, so	This was not determined by the search terms but by the team reviewing these articles in the scoping review. This is explained in closer detail in the scoping review which is cited in this review, and

health care services that don't involve service-users could have been included in results as well.	now also in the Method section (Masterson et al., 2022).
8. The inclusion/exclusion criteria reported were that of the prior scoping review, rather than that of the systematic review based on those original search results. If the presence of empirical data and measurements were required, as the 'initial screening' step would imply that they are, then they should be listed as inclusion criteria alongside the other inclusion criteria.	The inclusion/exclusion criteria for this systematic review are presented in the subheading Eligibility criteria and selection process for this systematic review .
9. It is also concerning that this step was taken separately from the following title/abstract screening step, which was conducted by two researchers and included additional checks to avoid bias.	As this review was using the articles included in the scoping review as a starting point, this step was an additional review of title and abstract for the inclusion/exclusion criteria for this review from the articles included in the scoping review.
10. In general, the authors need to clarify and distinguish the methods of the original scoping review from this systematic review.	To clarify the methods for the scoping review and this systematic review there are two subheadings in the method section: Information sources, eligibility criteria, search strategy and selection process for the scoping review and Eligibility criteria and selection process for this systematic review .
11. If the MMAT was used to assess quality, why was quality also assessed through the researchers discussing the data collection methodology and examples of the first findings with PhD students in the Samskapa research program? How did that impact the results? There doesn't seem to have been any significant quality assessment conducted.	One sentence is removed in the Method section ("To assess the risk of bias the researchers discussed the chosen data collection methodology and examples of the first findings with PhD students in the Samskapa research program") and another sentence is added: ("The discussions were intended to provide a fellow understanding of the meaning of the results for our future work in the Samskapa research program").
12. While the limitations mention the need for an update, they don't clarify whether the original scoping review search should be updated, or the subsection utilized for the systematic review.	This comment relates to comment No 3 above. For the moment, there is no plan to update the scoping review.
Reviewer Dr. Sanna Tuurnas, University of Vaasa	
13. The results show that there is a lack of common approaches to measuring outcomes of co-production. In my view this has a lot to do with the ambiguous nature of co-production concept itself. In this respect I was missing some critical discussion about the pluralism of the co-production concept and its impact on the outcomes or the results of study. As is well known, co-production can mean very different things from actual co-production to co-design. Co-production also takes place between different types of actors and agency (managers, professionals,	The definition, content and practice of co-production is demonstrably very varied. We share the belief that this diversity may have effects on co-production processes, the choice of measures and outcomes of co-production. However, based on how the analysis in our systematic review was conducted, it is difficult for us to comment on these effects more than we do. However, in the subheading Implications for future research we suggest that "An important implication for research concerns the need for closely described co-production processes, including clarifications of

politicians, patients, service users, citizens, family members). My comment therefore is, how are the type of co-production and the types of agency taken into an account in the research and how does this impact the way the process is measured, and what the outcomes are?	the activities, or process steps, that users participate in. Such clarity would provide better understanding of how co-production activities impact outcomes”. To avoid limitations on what stakeholders can co-produce in collaborations, we believe that the plurality has to be accepted – even embraced. In this systematic review, the balanced model is as suggested as one way to deal with the plurality, without restricting what and how stakeholders co-produce. Based on your comment, we have added three sentences in the Discussion. “(One plausible explanation for this lies in the ambiguous nature of the various ways in which co-production is conceptualised. Co-production is, and has been, defined and applied in multiple ways in health and social care [7]. Nonetheless, co-production was a positive experience for participants in our included studies and the findings provided learning important to enable a range of stakeholders to continuously engage in such processes”).
---	---